# Dental Auto Transplantation Success Rate Increases by Utilizing 3D Replicas

**DOI:** 10.3390/bioengineering10091058

**Published:** 2023-09-08

**Authors:** Peter Kizek, Marcel Riznic, Branislav Borza, Lubos Chromy, Karolina Kamila Glinska, Zuzana Kotulicova, Jozef Jendruch, Radovan Hudak, Marek Schnitzer

**Affiliations:** 1Department of Stomatology and Maxillofacial Surgery, UNLP, 04001 Kosice, Slovakia; peter.kizek@unlp.sk (P.K.); branislav.borza@upjs.sk (B.B.); karolina.kamila.glinska@upjs.sk (K.K.G.); zuzana.kotulicova8@gmail.com (Z.K.); jendruch.jozef@gmail.com (J.J.); 2Department of Biomedical Engineering and Measurement, Faculty of Engineering, TUKE, 04200 Kosice, Slovakia; lubos.chromy@tuke.sk (L.C.); radovan.hudak@tuke.sk (R.H.); marek.schnitzer@enbicore.eu (M.S.)

**Keywords:** dental autotransplantation, 3D printed replica, additive manufacturing

## Abstract

Dental autotransplantation is an effective alternative to conventional dental treatment, and it involves removing a tooth and repositioning it in a new position within the same patient. Although this procedure might pose more intraoperative challenges, it provides a great solution for replacing missing teeth or aiding difficult eruption in young patients. This prospective method is also advocated as a use of treatment for unrestorable teeth. The success rates of autotransplantation cases with and without replicas were compared in a retrospective analysis of the data. By reducing donor tooth manipulation and ensuring a proper fit and positioning in the recipient socket, replicas significantly increased success rates of the procedure. CBCT scans were used to collect data. Data exported to the Mimics system were then processed in order to achieve a model of the donor tooth. Additive manufacturing technology was used to create the replicas. Specialized biocompatible material was used. Details of the replantation site and the donor tooth requirements were described, as well as the step-by-step surgical technique. For the best results, variables, like patient selection, surgical technique, and long-term monitoring, were found to be essential. The study highlights the significance of dental professionals and biomedical engineering staff working together to develop standard operating procedures and achieve predictable outcomes in autotransplantation procedures. The results suggest that 3D printed replicas could be a useful tool for improving the effectiveness and success of dental autotransplantation.

## 1. Introduction

Dental autotransplantation is the removal of a tooth from its original location in order to place it in an edentulous zone, whether it is a post-extraction socket or a surgically prepared socket [1]. This therapeutic option widens possibilities of dental rehabilitation of missing or lost teeth, beyond the use of prosthetic or standard surgical treatment, such as use of dental implants. Success of the procedure depends on certain variables, such as preservation of recipient site, successful atraumatic extraction of the donor tooth, as well as the need of endodontic treatment of the donor tooth. Minimal manipulation of the donor tooth perioperatively makes the process especially difficult when it comes to positioning of tooth in the correct position. This is necessary to preserve periodontal ligaments (PDL) which are responsible for correct healing after the placement into the recipient bed [2,3]. In recent years, the use of digital templates in autotransplantation has been widely debated [4,5,6]. Typically, 3D printed replicas are made by using computer-aided design (CAD) rapid prototyping (RP). Noninvasive optical scanning is currently expanding in many clinical dental applications, including fabrication of customized appliances for patients with craniofacial disorders [7]. The use of dental replicas may be valuable in visualizing and correcting the position of the donor tooth in the recipient socket, without the need of manipulation with a donor tooth. It additionally provides the benefit of shortening the extra-alveolar time of the donor tooth [4,8]. The extra-alveolar time is the time between the extraction of the donor tooth and its insertion into the recipient site. The success rate of autotransplantation procedures has been shown to increase when the extra-alveolar time is kept to a minimum [4]. The purpose of this work is to retrospectively compare the success rate of the autotransplantation with application of the 3D printed replica. The manufacturing and application of dental replicas in actual clinical settings are described. This study contributes to a more comprehensive understanding of the role and the process of manufacturing 3D technology in enhancing the efficacy of dental autotransplantation procedures. Additionally, this study investigates the prospective implications of guided autotransplantation beyond its surgical aspects. By analyzing the outcomes of the procedure, such as post-operative complications and endodontic treatment need, the study aims to provide a comprehensive perspective on the viability and sustainability of guided autotransplantation. The analysis of clinical and radiographic data contributes to a nuanced evaluation of the effect of using 3D replicas on the procedure’s overall success.

Ultimately, as dental autotransplantation gains prominence as a viable treatment option, the incorporation of cutting-edge technologies, such as 3D printed reproductions, has the potential to transform the landscape of dental rehabilitation.

## 2. Materials and Methods

A retrospective cohort study based on STROBE statement eligibility criteria was carried out to investigate the efficacy of autologous transplantation procedures with and without the use of replicas. A demonstration of the manufacturing process for replicas used intraoperatively is described.

### 2.1. Collection of Data of Autotransplantation Success Rate

A retrospective study was conducted in a structured manner and obtained ethical commission approval, number 2023/EK/06035, prior to implementation. Data collection on the autotransplantations carried out between 2019 and 2022 was studied.

All accessible information on the autotransplantation technique has been compiled. Patients were selected for the surgery based on their general health, dental hygiene, and willingness to comply. The significance of dispensing with weekly follow-ups throughout the first month of the healing process was significant. Case-by-case, patients were selected based on the availability of the donor tooth, the possibility of its extraction without destroying it if it was an ectopically impacted donor, and the size and quality of the recipient bone. These requirements, as outlined in Table 1, are crucial for determining if autotransplantation may be scheduled.

The leading indication in younger patients was the ectopic impaction of upper canines, where appropriate orthodontic treatment was already partly taken at the time of the autotransplantation, mainly involving the space preparation for the insertion of the autotransplanted donor tooth in case it was agenized. Alternatively, the state of the preexisting bone in the area of transplantation of the carious tooth is a clear contraindication, as is the presence of large apically present inflammation or a lack of the bone in the crestal area, as the retention of the tooth could be impaired. The autologous bone was used in cases where it could be received from the area of the transplant. The use of xenografic bone graft material, although possible, was avoided in order to avoid introducing another variable into the process of the healing, as its position in the aid of current healing is unclear [9].

The application of the approach to different teeth and a comprehensive assessment of all advantages and disadvantages in comparison to other treatment procedures was the principal variation. The primary variables were the patient’s age and the indication of the autotransplantation.

Diagnostic criteria included CBCT scans of the donor tooth’s architecture and closeness to important tissues, as well as a comprehensive medical history.

The collected data were split into two study groups. A replica was used for autotransplantation on 10 patients in Group 1 (year 2020–2021) who ranged in age from 13 to 34 (mean 16.2 years). Group 2 (year 2019–2020) consisted of 14 patients, ages 15 to 54 (mean 21.9 years), who underwent surgery without the use of a replica. All of the procedures were performed by the same surgeon. The method of using a 3D replica has been set up in our department in 2020 and, since then, all cases of tooth autotransplantation have been solved using this technique (Group 1).

The majority of autotransplantations were carried out in patients undergoing orthodontic treatment with ectopic impaction of maxillary canines. The second most frequent indication was the autotransplantation of the first mandibular molar. Certain optimal donor tooth and recipient site qualities for autotransplantation must be met in both groups (Table 1).

Both groups were observed for 14 months on average, with the time of observation ranging from 12 to 36 months. Patient preparation included an informed consent signature and information about other protentional solutions and risks associated with the procedure.

Tooth vitality was checked by the cold test after 6 months and 12 months since the autotransplantation procedure. In case the tooth vitality was negative, an endodontic procedure was carried out within two weeks of the surgery, as recommended by many authors [10,11]. In our experience, the majority of patients did not need endodontic treatment due to positive pulp vitality preservation. The concept of not performing RCT after autotransplantation as a standard technique has been considered in the literature [12,13]. However, many studies have described performing the endodontic treatment as a standard procedure within a two-week period of autotransplanting [11].

The following criteria have been established to determine whether a procedure was successful after a year from the time of the procedure: there must be no pain, no mobility, and no periapical resorption or pathologies. The study has examined various data points, including extra-alveolar time, ankylosis rate, the necessity of endodontic treatment, and procedure failure rates, both with and without the use of a replica.

### 2.2. Methodology of CBCT Data Processing

A cone beam computed tomography (CBCT) scan (0.4 mm) of the upper and lower jaw was obtained. The scanned data were imported into Mimics software (V21; Materialise; Leuven, Belgium) in DICOM format. Three main planes—axial, sagittal, and coronal—were used to display CBCT scans. To segment a certain tooth, the dental tissue selection needs to be defined using the Threshold function, which corresponds to various shades of gray in HU (Figure 1). To simplify the segmentation, the area of interest (lower right wisdom tooth) can be de-limited. Different HU ranges are also defined for the teeth under the Threshold function (tooth (CBCT); min = 1200 HU, max = 3071 HU).

Using the edit masks and the erase (lasso) function, individual tissues were separated from the selected tooth in individual sections and in the sagittal and axial planes. After a manual segmentation, the Region grow function was selected and a new mask (Figure 2) was created and separated the segmented tooth from the remaining superfluous tissues defined under the Threshold mask (Figure 1). The Calculate component function was picked from the provided choices and set to Optimal to produce a 3D model of the tooth. Following the creation of the 3D model, the Smooth function was utilized, with a smoothing value of 0.8 and a repeat of 3. The developed model was exported to the standard triangle language (STL) model required for 3D printing (Figure 3).

### 2.3. Model Preparation and Production

The replicas were created using additive manufacturing technology. An EnvisionTEC Vida printer was used, featuring DLP (digital light processing) technology. The EnvisionTEC Vida operates on the light projection (DPL) concept. It supports the use of a variety of materials, allowing for 3D printing of a broad range of dental, orthodontic, hearing, and other applications. E-Guard material (EnvisionTEC, Germany) was utilized to create the replicas, which is a biocompatible translucent material often used in digital dentistry or orthodontic labs in practice.

The STL model was imported into the PreForm software (Formlabs, Somerville, MA, USA), in which the replica was prepared for printing. After importing, the main parameters for production were set, and the material used and the layer thickness were chosen. The correct position of the model was determined, and the support structure was generated (Figure 3). The prepared model was then sent to the 3D printer, and the production began (Figure 4).

### 2.4. Surgical Methods

If a replica tooth was used beforehand, it was prepared in accordance with the principles of asepsis (according to instructions for use—E-guard: disinfection with 70% ethanol solution before use). The autotransplantation can be performed with local or general anesthesia depending on the position of the tooth and the difficulty of the surgery, as well as the compliance of the patient. Before the procedure, the oral cavity and surrounding tissues are disinfected as part of the preparation for the autoimplantation site. The extraction of the tooth or raising a flap in edentulous area should consist of minimal trauma, The recipient socket was prepared with abundant saline irrigation, a surgical contra-angled handpiece with an implant surgical kit, using an ascending drill width to achieve a proper socket size. A paralleling pin (a) and sanitized replica were used to check the socket angulation and depth of the implant bed (b) (Figure 5). An occlusal contact of the replica and the antagonist tooth was checked using the articulation paper—no contact should be registered. After reaching the proper position, the replica was left in the new socket for prevention of bone bleeding and to minimize the contact of the bone with the external environment. The replica was left in the socket until the final placing of the donor tooth.

The donor tooth was extracted utilizing a minimally traumatic technique Figure 6a, which included raising the surgical flap in case of its impaction or releasing of PDL with scalpel before extraction with forceps. Bone removal from the area of the root surface had to be avoided due to the risk of damaging the PDL. After confirmation of a satisfactory position, the donor tooth was finally removed from its socket. A leukocyte- and platelet-rich fibrin (L-PRF) concentrate previously prepared by a standard protocol was placed into the neo-alveolus, and the donor tooth was gently placed into its position. A stopwatch was used to measure the time between the extraction of the donor tooth and its insertion into the new socket. Special care was taken to insert the donor tooth in a single attempt, as previously simulated by the replica. The donor tooth was quickly inserted in the new area and fastened with a suture. In cases where the patient undergoes an orthodontic treatment Figure 6b with a fixed appliance, the replanted tooth can be fixed with brackets. The donor tooth was checked by the articulation paper once more.

Post-surgical instructions included a soft diet for one month and to avoid mobilizing or touching the “new tooth” as much as possible. Medical prescription included antibiotics: metronidazole 500 mg (tablet 3 times a day for 7 days) and amoxicillin 500 mg (capsule 3 times a day for 7 days), non-steroidal anti-inflammatory drug: diclofenac potassium 50 mg (tablet 3 times a day for 3 days), and mouthwash with chlorhexidine gluconate 0.12% (twice a day for 10 days). The patient must be compliant and adhere to post-operative instructions. Stitches can be removed 2 weeks post-surgery.

## 3. Results

Success Rate Study of Use of Replica

The extra-alveolar time of the tooth with replica averaged 12 s, whereas the group without replica averaged 5.5 min. Three out of fourteen of the Group 2 patients showed radiological signs of ankylosis in the autotransplantation site after six months post-op. Three patients required endodontic treatment in Group 1, and four patients did in Group 2.

Through clinical and radiographic examination, the research found that autotransplanted teeth in Group 1 had a success rate of 90%, while Group 2 had a success rate of 71.4%. (Table 2).

## 4. Discussion

Dental autologous transplantation represents an intriguing choice of treatment in patients with defects caused by a missing or decayed tooth.

The benefits of this technique include quick tooth replacement, effective transplanted tooth adaptability to growth changes, and a typical orthodontic force response. These characteristics are crucial for children and young adults, for whom titanium dental implants are typically not recommended [14].

The major discrepancy between the average age of Group 1 and Group 2 was due to the fact that in Group 1 there was a predominance of patients with an indication for the transplantation of an impacted tooth (mostly younger patients), and in Group 2 there was a predominance of replacement of the first molar with a wisdom tooth (patients who were older in age).

Replicas were employed particularly in situations involving impacted canines, primarily among younger patients. These cases often entail a prolonged extraction process and the preparation of a new socket (neo-alveolus). In contrast, the extraction of third molars utilized as donor teeth, contributing to the creation of a neo-alveolus or being placed into a preexisting socket as part of an immediate autotransplantation procedure in space of first mandibular molar, was more common in older patients. This distinction in the extraction process and patient age gives rise to the observed mean age difference in both groups.

Both healing of the periodontal ligament tissue and pulp is crucial for the successful preservation of the donor tooth. The establishment of the healed tissue with vascularized connective fibers has been observed on the 30th day from the time of autotransplantation [15]. The vascular supply present near to pulp is crucial for the pulp vitality preservation. This study, however, has only been applied in cases of transplantation where the incomplete root developed, as dental papilla is still present in that case.

According to certain authors, if replanted teeth have their root development completed, the process of successful revascularization of the pulp is unlikely. The aforementioned endodontic treatment is, therefore, deemed necessary and should be carried out before the procedure if the donor tooth is accessible, or in a two-week period after the surgery [11]. Nevertheless, in the experience of the operator this was not often necessary, as the endodontic treatment was far less required, and revascularization of the donor tooth has been noted [16].

In cases of an immature root apex, Machado et al. identified positive pulp reaction after 2–4 months in 81% of cases [17]. In situations where the recipient site was too large for the replanted tooth, augmentation with either a xenograft or autologous bone material is necessary [18].

Using computer-aided planning, including CT scans and replicas, can ensure that the recipient site is optimal for replantation. Use of CT or CBCT on such an application is still superior to the accuracy offered by optical scanning capabilities [19]. Confirming the correct size and position of the donor tooth with the use of a replica can lay groundwork for a fast and successful process. This, combined with optimal selection of patients and surgical technique as shown in article, can secure predictable success when applying this technique. The occurrence of ectopic tooth eruption may also serve as an indication for the usage of autotransplantation. Typically, teeth that are in ectopic positions are surgically exposed, followed by the application of orthodontic traction [20]. Maxillary canines are crucial for dental and facial aesthetics as well as correct occlusion. When surgical techniques of exposure are inadequate, autotransplantation should be explored. Correction of the position of severely ectopic maxillary canines, which are found in approximately 2% of the population, can pose a significant challenge for conventional orthodontic techniques [21,22]. Hence, the use of replicas during autotransplantation of ectopic canines to achieve a more natural alignment may offer a streamlined and expedited approach to treatment. Furthermore, considering the occurrence of caries defects in first molar is high, as it is the first tooth to erupt, it often leads to unrestorable teeth. Young patients who have first permanent molar loss may develop irregular occlusion as a consequence of tooth migration and uneven jaw development [23]. Therefore, the goal of such a patient’s therapy should be to keep the gap left by their missing teeth without altering their developing jaw [24]. Often a wisdom tooth may serve as a donor in this instance. This is because it is the last tooth to be developed, and often in the time of need of autotransplantation its apex is still open, which increases the possibility of preservation of vitality [25,26].

The success rate of autotransplantation of teeth using a 3D replica seems high and increasingly available [27,28,29]. However, it is necessary to maintain the rate of long-term survival of the tooth with respect for its function, occlusion, and aesthetic result. The accuracy of tooth autotransplantation, therefore, requires long-term monitoring of patients radiographically as well as clinically. The question arises whether the short time of the replantation process, observed also with the help of 3D technology, does not contribute to the vitality of the tooth, which can persist for a long time without the risk of progressive resorption and, thus, ankylosis and subsequent endodontics. Several authors, representing a variety of perspectives, have proposed procedure standardization [30,31].

The simulation of the insertion to the neo-alveolus enables the procedure to be undertaken in one step as a pathway of insertion is cleared and confirmed by previous alignment of the replica. In addition, a dimension of the recipient site, including an appropriate fit of bone and root of autologous transplant is confirmed to ensure a more favorable prognosis for the donor tooth. However, the process of digitizing clinical data, transmitting the digital planning, and carrying out the autotransplantation might result in accumulated dimensional disturbances [32,33]. This might result in a difference between the anticipated position of the donor tooth and the actual fit of the newly allocated tooth.

Success is relies on close cooperation between the doctor, the biomedical engineering staff, and the entire operating team.

## 5. Conclusions

This study demonstrates the details of both the surgical procedures and technicalities associated with the replica preparation. The proposed methodology may make this rehabilitation technique more accessible for operators to apply in daily practice. The primary limitation of this research is that it is a retrospective examination of data that compensates for restricted points of comparison based on data acquired in patients’ records. Other limitations are a small study group and a relatively short follow-up period. Later in the research, the decision to use a replica was made based on the availability of the method. The diverse phases of tooth development, interchangeable tooth types, and site of autotransplantation compensate for characteristics not accounted for in the research.

The study provided evidence that the utilization of three-dimensional replicas effectively decreased the duration of extra-alveolar time, thereby mitigating the potential adverse effects on tooth vitality and long-term results. The utilization of this technology facilitates controlled insertion trajectory, thereby ensuring a streamlined, single-step procedure and diminishing the necessity for repeated attempts in accommodating the transplanted tooth.

In addition, the utilization of 3D replicas played a crucial role in facilitating the preservation of cells on the root surface, thereby promoting optimal healing of the transplanted tooth.

In a wider framework, dental autologous transplantation continues to be a valuable therapeutic alternative for individuals who have missing or decayed teeth, especially in cases where titanium dental implants may not be appropriate, such as in younger patients. The study’s findings demonstrate the potential for achieving positive outcomes in tooth transplantation procedures, particularly when 3D technology is utilized.

In order to advance the field, it is imperative to conduct additional research and establish standardized protocols to enhance the utilization of 3D technology in dental autotransplantation. The implementation of long-term radiographic and clinical monitoring of patients will yield significant insights into the durability of the documented success rates and the effects of employing 3D replicas on tooth vitality and long-term functionality.

## Figures and Tables

**Figure 1 bioengineering-10-01058-f001:**
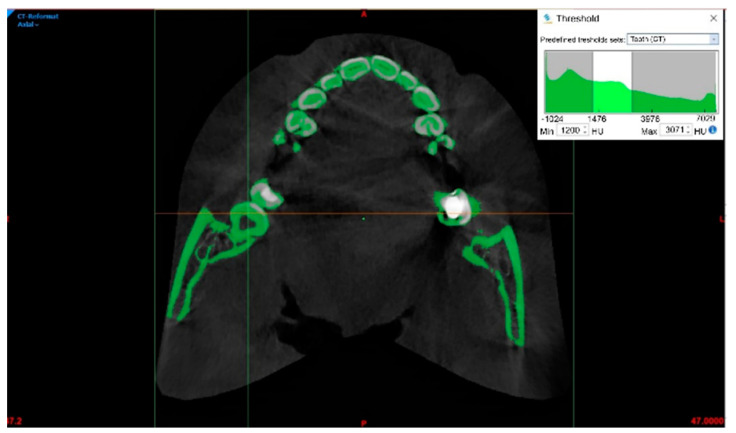
Definition of the threshold donor tooth shown by the arrow.

**Figure 2 bioengineering-10-01058-f002:**
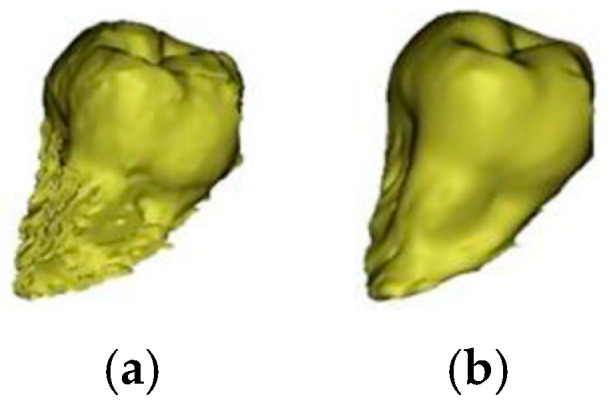
3D model of the tooth (**a**) before the smoothing of the contours and (**b**) after the smoothing of the contours.

**Figure 3 bioengineering-10-01058-f003:**
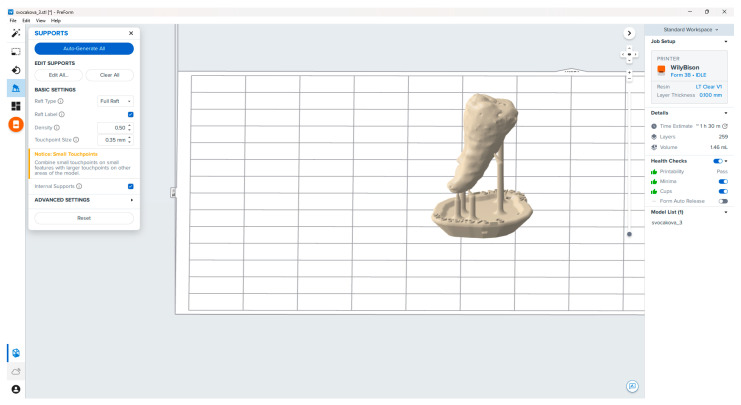
Preparation of the model for printing.

**Figure 4 bioengineering-10-01058-f004:**
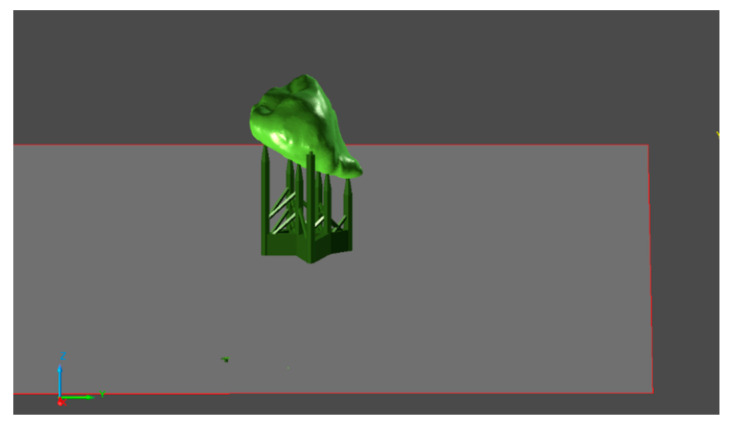
Determination of the position of the model and application of the support structure.

**Figure 5 bioengineering-10-01058-f005:**
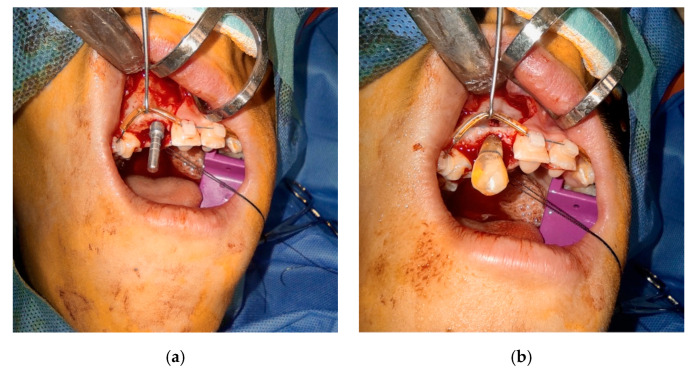
Confirmation of the size and angulation of the new socket (**a**) with paralleling pin and (**b**) simulation of the pathway of the tooth insertion with the replica.

**Figure 6 bioengineering-10-01058-f006:**
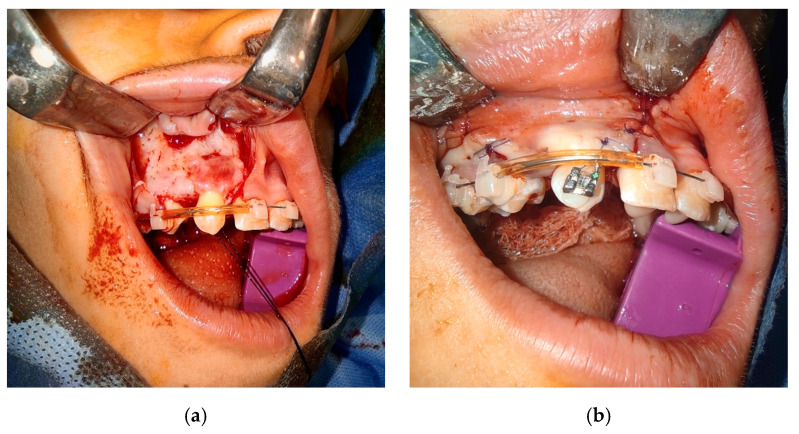
Insertion of the donor tooth into a recipient site (**a**) and securing of the donor tooth with orthodontic bracket (**b**).

**Table 1 bioengineering-10-01058-t001:** Description of the requirements for the replantation site and donor tooth.

Recipient site	**Sufficient mesio-distal width ≥ 8 mm for canine and ≥11 mm for wisdom tooth**
	No chronic/acute infection
	Healthy surrounding soft tissues
	Appropriate bone level and width of alveolar bone
Donor tooth	No occlusal interference
	Suitable root morphology
	Healthy bone surrounding donor tooth

**Table 2 bioengineering-10-01058-t002:** Comparison of auto transplantation results with and without replicas.

Autotransplantation Type	Number of Patients	Age (Mean ± SD)	Age Range (Years)	Extra-Alveolar Time (Seconds)	Success Rate	Ankylosis Rate	Need for Endodontic Treatment
Group 1	10	16.2	13–34	12	90%	0%	30%
Group 2	14	21.9	15–54	330	71.4%	21.4%	35%

## Data Availability

The data presented in this study are available on request from the corresponding author. The data are not publicly available due to the ethical reasons.

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
