# Peer review of "Dental Auto Transplantation Success Rate Increases by Utilizing 3D Replicas"

_bioengineering, 2023, doi:10.3390/bioengineering10091058_

Round 1

Reviewer 1 Report

The authors present their results of teeth autotransplantation with and without utilizing 3D Replicas. The procedure is becoming increasingly important due to its benefits for the patient.

There is a review available on this topic (Verweij ,2017), that was cited in the manuscript. Recently a guide was published (Tsukiboshi, M, Tsukiboshi, C, Levin, L. A step-by step guide for auto transplantation of teeth. Dental Traumatology. 2023; 39(Suppl. 1): 70–80.) which should also be included in the manuscript.

The benefit of the manuscript is the description of the replica deigning in the SW, although it should be revised (see specific comments below).

The English is poor, and language should be revised.

The structure of the manuscript is poor; the manuscript should be reorganized. (see specific comments below)

Abstract

It should be structured and should not include citations

Introduction

The purpose (aim) is too long. Please shorten.

M&M

Subtitle “2.2. Methology of CT Data Processing” should be changed in “2.2. Methodology of CBCT Data Processing

Replace CT with CBCT through the text.

CBCT images are not using HU.

How did you “sanitize” 3D replicas? Sterilization in autoclave should be used.

Results

Shorten the Results section.

The description of the surgical procedure should be moved under M&M section.

The last paragraph, describing success criteria, should be moved to M&M section.

Language should be improved.

Author Response

Please find the reply to your review in the attachment

Reviewer 2 Report

This paper aims to present the outcomes of a retrospective investigation into the success rates of autotransplantation using 3D printed dental replicas. Additionally, it outlines the manufacturing process of the dental replicas that were utilized in the collaborative effort.

Currently, the classification of this paper as a randomized controlled trial (RCT), retrospective cohort study, or technical note is challenging. If it aligns with an RCT, the authors are encouraged to adhere to the CONSORT guidelines for reporting their work. Should it fall under the category of a retrospective study, the authors are advised to employ the STROBE guidelines for cohort studies. If it is more fitting as a technical note, the authors should provide a more explicit delineation of how their protocol enhances existing protocols, as listed in the references (https://pubmed.ncbi.nlm.nih.gov/?term=3d+autotransplant+tooth).

To enhance clarity, the following points need further elucidation:

1. The method used to assign patients to either group 1 or 2.

2. Specification of the teeth that underwent transplantation.

3. The number of transplanted teeth necessitating orthodontic intervention.

4. An explanation of the implications stemming from the mean age difference between group 1 and 2.

Providing this additional information will contribute to a more comprehensive understanding of the study's methodology and findings.

Author Response

(The authors gave the same response as above.)

Reviewer 3 Report

Auto transportation is an emerging area of research in dental studies and requires additional studies to maturity. The article discusses the subject by examining the success rates of auto-transplantation through illustration with the 3D printed replica. Given the above perspective, this article is potentially useful to the dental community and has the chance of extensive referencing if polished for publication with the following suggestions:

1. Though the statements referenced in the abstract may be retained, the reference numbers should be removed. References are not allowed in abstracts. If the authors have copied verbatim from sources, the text should be paraphrased. This concerns references [1] and [2]. Thus introduction should start with [1].

2. The novelty of the article should be defined clearly.

3. Conclusion section must be included.

4. The starting statement of the second to the last paragraph of the discussion section must be re-written; it is grammatically ineffective.

5. Future aspects and limitations of the study should be included in the conclusion section.

Author Response

(The authors gave the same response as above.)

Round 2

Reviewer 2 Report

The revised version of this manuscript has addressed most of my previous concerns, which is appreciated. However, the clarity of the study design still requires improvement. Additionally, the Methods section is heavily focused on describing the procedure, but lacks sufficient detail on patient selection and variables measurement. It's crucial that the authors adhere to the STROBE guidelines to enhance the reproducibility of their methods.

Author Response

Thank you very much for your feedback. The section Methods has been improved by patient selection and variable measurements section.